# OpenReview forum: "SalUn: Empowering Machine Unlearning via Gradient-based Weight Saliency in Both Image Classification and Generation"
_ICLR.cc/2024/Conference — ICLR 2024 spotlight_

### Official Review · Reviewer_vf6N · 2023-10-29

**Soundness:** 3 good
**Presentation:** 2 fair
**Contribution:** 3 good
**Rating:** 8
**Confidence:** 2

**Summary:**

The authors propose a method called Saliency Unlearning (SalUn) that aims to improve the efficiency and effectiveness of MU in both image classification and generation tasks.

**Strengths:**

1. The paper is well-structured and self-contained.
2. The experiments are thorough, comparing SalUn with multiple baselines in different scenarios.
3. The paper broadly investigates previous methods, discusses their limitations, and proposes a versatile method to alleviate them.

**Weaknesses:**

1. The quality of many generated images is bad.
2. The unlearning of the generation task is only conducted with one version of the Latent Diffusion Model. Considering that many personalized models trained with Dreambooth or more powerful models like SDXL, DALLE, and Imagen are available, they should be further tested to verify their universal effectiveness. Also, the generation of images should be repeated with different seeds to avoid cherry-picking.

**Questions:**

I have to say, although relevant, I have limited knowledge about machine unlearning, and I will consider the opinions of other reviewers to make further decisions.

**Details Of Ethics Concerns:**

Some images shown in this paper might be offensive. I am not sure whether they are acceptable. There are many horrible faces and bodies in Fig. 6. I think it's better to avoid examples of famous public people like Putin in P3 (Fig. 6).

---

> ### Author Response · Authors · 2023-11-19
> **Response to Reviewer vf6N**
>
> We express our sincere gratitude to Reviewer vf6N for the recognition of the completeness of our experimental work. And greatly appreciate for proposing insightful comments regarding the generative model perspective. Below, we provide our responses to the comments, denoted by **[W]** for weaknesses and **[E]** for ethics concerns.
>
> **Response to W1 regarding low generation quality:**
>
> This is a valuable comment. Yes, MU could trigger a possible tradeoff between unlearning effectiveness (*i.e.* UA) and preserved generation quality (*i.e.* FID). This is an open challenge that remains in the field: How to strike the optimal tradeoff between these two factors [R1]. Table 2 shows such as tradeoff to some extent, where our methods achieve a much better unlearning performance than the baselines ESD and FMN while suffering a slight generation quality drop in FID. We will clearly point out this tradeoff and provide more discussions in the revision.
>
> > [R1] Jia, Jinghan, et al. "Model Sparsity Can Simplify Machine Unlearning." Thirty-seventh Conference on Neural Information Processing Systems. 2023.
>
> **Response to W2 regarding focusing on one version of the Latent Diffusion Model:**
>
> Thank you so much for your suggestions!
>
> Our evaluation is focused on SD (stable diffusion) due to the fact that the evaluation benchmark I2P (for NSFW generation) and its testing framework were specifically designed around this latent diffusion model [R2]. Therefore, it serves as the most established testbed for assessing the performance of our unlearning method, allowing for fair comparisons with other baseline approaches such as ESD [R3] and FMN [R4] that use I2P. However, inspired by the reviewer’s suggestion, in the revision we will conduct a sanity check for I2P on other versions of latent diffusion models and implement our method to remove its harmful influence or other NSFW-related prompts/concepts.
>
> We want to express our gratitude for your suggestion to consider multiple seeds for image generation. It is important to note that we have already included the generation results from multiple seeds in Figures A1, A2, and A3. We are committed to providing additional results from various random seeds to ensure a comprehensive and robust evaluation of our method. Your feedback has been invaluable in enhancing the completeness of our research.
>
> To clarify, in the NSFW-removal evaluation, we are using an I2P benchmark, which contains over 4.5k seed and text prompt pairs that are likely to produce inappropriate content.   In this scenario, the seeds are fixed with given prompts. Thus, following I2P, we did not break the pairing between the fixed seed and the fixed text prompt in Figure 6. The same strategy was also adopted in the baseline ESD for NSFW-removal evaluation.
>
> > [R2] Schramowski, Patrick, et al. "Safe latent diffusion: Mitigating inappropriate degeneration in diffusion models." Proceedings of the IEEE/CVF Conference on Computer Vision and Pattern Recognition. 2023.
> >
> > [R3] Gandikota, Rohit, et al. "Erasing concepts from diffusion models." arXiv preprint arXiv:2303.07345 (2023).
> >
> > [R4] Zhang, Eric, et al. "Forget-me-not: Learning to forget in text-to-image diffusion models." arXiv preprint arXiv:2303.17591 (2023).
>
> **Response to E on offensive image generation:**
>
> We acknowledge the ethical importance of your suggestion. In the revised manuscript, we will apply mosaic and blurring on generated images to prevent the dissemination of offensive content.
>
> ------
>
> Last but not least, we would like to underscore our contributions and the solidity of our approach to machine unlearning, as highlighted by other reviewers:
>
> **Reviewer miKP**'s comments acknowledged the significance of our work:
> > "Significance: the task is very relevant: machine unlearning for generative models, a very important topic to prevent harmful content generation, was lacking an effective solution."
>
> As well, **Reviewer XDsn** also indicated our novelty:
> > "SalUn introduces a novel gradient-based weight saliency approach, a significant departure from existing MU methods."
>
> **Reviewer TAAE** also acknowledged our contributions:
> > "The proposed method is simple, intuitive, and extremely effective according to the experiments in the paper."
> >
> > "Image generation is a great benchmark for these models, kudos to the authors for including experiments with diffusion models."
>
> In light of the above, we hope that Reviewer vf6N can now better understand our work and its novelty.

---

> ### Author Response · Authors · 2023-11-22
> **Reminder on follow-up discussion (one day left before rebuttal ends)**
>
> Dear Reviewer vf6N,
>
> We sincerely appreciate the time and effort you've invested in reviewing our manuscript. Your expertise and dedication to the review process are truly invaluable to us.
>
> As the discussion phase draws to a close in one day, we kindly request you to provide your feedback on our rebuttal. Your insights are of immense importance to us, and we look forward to any additional comments you may have. If our replies have addressed your concerns, we would be thankful for your recognition of this. Should there be any points requiring further discussion or explanation, please feel free to contact us. Furthermore, we are fully prepared to engage more to enhance our work during this pivotal stage.
>
> Best regards,
>
> Authors

---

> > ### Comment · Reviewer_vf6N · 2023-11-22
> >
> > Thank you for your feedback. I will upgrade my rate to accept.

---

> > > ### Author Response · Authors · 2023-11-22
> > > **Thank you!**
> > >
> > > Dear Reviewer vf6N,
> > >
> > > Thank you for your swift response and for acknowledging our efforts in addressing your questions. We are pleased to hear that our responses have been satisfactory.
> > >
> > > Thanks,
> > >
> > > Authors

---

### Official Review · Reviewer_TAAE · 2023-11-07

**Soundness:** 3 good
**Presentation:** 3 good
**Contribution:** 3 good
**Rating:** 6
**Confidence:** 4

**Summary:**

This paper introduces a novel approach called "Saliency Unlearning" (SalUn) in the context of machine unlearning (MU) to improve the accuracy, stability, and cross-domain applicability of unlearning data from AI models. SalUn focuses on specific model weights, similar to input saliency in model explanation, and effectively erases the influence of forgetting data, classes, or concepts in both image classification and generation. SalUn outperforms existing methods and achieves high unlearning accuracy, even in challenging scenarios like random data forgetting and harmful image prevention.

**Strengths:**

- Machine unlearning is critical, especially now that large models trained on the whole internet are available to everyone
- The proposed method is simple, intuitive, and extremely effective according to the experiments in the paper
- Image generation is a great benchmark for these models, kudos to the authors for including experiments with diffusion models
- The paper is well written, and the figures are intuitive. The whole thing feels polished and appropriate for the venue.

**Weaknesses:**

- It would have been better to disregard classification experiments and have language generation experiments instead. This would also elucidate possible challenges when trying to unlearn sequential knowledge where the classification of each slot (token) depends on previous tokens as well. This might impact the proposed method, since it is harder to pinpoint the weights that caused the unwanted behavior. Also, the random labeling approach for estimating saliency would also be suboptimal, because many tokens like articles and prepositions might be used in harmful as well as harmless sentences.
- Experiments with zero-shot classification (CLIP) would have also been preferable wrt pure classification
- There are many methods for weight saliency estimation, it would have been nice if the authors had ablated this component using different estimation methods. See continual learning literature.
- Limitations of the proposed method should also be addressed

**Questions:**

See weaknesses

---

> ### Author Response · Authors · 2023-11-19
> **Response to Reviewer TAAE (1/2)**
>
> We thank Reviewer TAAE for acknowledging the contributions, soundness, and presentation quality of our paper. We also greatly appreciate Reviewer TAAE for proposing the very insightful questions. Below, we provide our responses to the comments, denoted by **[W]** for weaknesses and **[Q]** for questions.
>
> **Response to W1 on disregarding classification and considering language generation tasks:**
>
> We appreciate your insightful question! It is worth noting that unlearning for image classification tasks holds an important role within our work. Our unlearning approach has primarily been tailored for vision-related tasks rather than language-related ones.
>
> 1. **Reason for incorporating image classification tasks:** MU has received considerable attention and extensive study within the realm of image classification tasks. A multitude of MU methods have been developed for this purpose. However, it remains skeptical that none of the existing works applied these methods to image generation tasks, which carries a significant need to avoid generating content associated with NSFW (Not Safe For Work) concepts. To delve deeper into this, we conducted an initial examination of existing MU approaches applied to image classification in Section 4. During this investigation, we identified certain limitations inherent to these methods, *i.e.*, the lack of stability and task generality. Building upon these results, we introduced the concept of weight saliency for unlearning. We aim to maintain the effectiveness of MU across both classification and generation tasks.
> 2. **Divergence between vision models and language models:** We appreciate the reviewer's suggestion regarding the potential of MU for language generation, which is indeed an intriguing avenue for future research. However, we must acknowledge that this is beyond the scope of our current work. The divergence between vision models and language models introduces unique challenges, such as the intricate investigation required for defining structural units for saliency computation in language models, e.g., attention heads or weights/columns in feedforward network modules. Moreover, as pointed out by the reviewer, language models present other complexities in the selection of unlearning saliency loss functions. Furthermore, our awareness of machine unlearning works for language models was limited [R1, R2, R3]. Encouraged by the reviewer's feedback, we plan to incorporate these relevant works into our related work section and acknowledge the potential of exploring MU in language generation as a direction for future research. Thank you for your valuable insights.
>
> > [R1] Pawelczyk, Martin, Seth Neel, and Himabindu Lakkaraju. "In-Context Unlearning: Language Models as Few Shot Unlearners." arXiv preprint arXiv:2310.07579 (2023).
> >
> > [R2] Yao, Yuanshun, Xiaojun Xu, and Yang Liu. "Large Language Model Unlearning." arXiv preprint arXiv:2310.10683 (2023).
> >
> > [R3] Eldan, Ronen, and Mark Russinovich. "Who's Harry Potter? Approximate Unlearning in LLMs." arXiv preprint arXiv:2310.02238 (2023).

---

> ### Author Response · Authors · 2023-11-19
> **Response to Reviewer TAAE (2/2)**
>
> **Response to W2 regarding zero-shot classification (CLIP):**
>
> We acknowledge that zero-shot classification is indeed an interesting setting for unlearning. Building upon the reviewer's suggestion, we carried out an additional experiment involving a pre-trained ResNet-50-based CLIP model (see **Table R4**). In this experiment, we aimed to assess the unlearning ability of CLIP to "forget" the influence of a particular CIFAR-10 image class (*i.e.*, the target class for unlearning). Given the zero-shot classification constraint, we assume that the unlearner only has access to the forgetting dataset, consisting of CIFAR-10 data points belonging to the class designated for unlearning. In this context, the suitable unlearning method is GA (gradient ascent) which only relies on the forgetting dataset [R4], and our proposal becomes the saliency-guided GA (like GA + $m_S$ in Table A1). The unlearning evaluation follows Table A1, where **UA** indicates more effective unlearning, and the higher **TA** signifies better generalization. As we can see, our approach outperforms GA in both unlearning effectiveness and preserved zero-shot classification ability. Meanwhile, we also observed that the GA-based unlearning seems to inevitably hamper the zero-shot classification ability at testing time. We believe that unlearning for zero-shot classification presents its own distinct set of challenges. This is primarily because there can be a substantial disconnect or mismatch between the training dataset and the testing dataset in this context. This is a very insightful question. Thanks!
>
> **Table R4**: MU Performance of zero-shot classification (CLIP), for forgetting within a single class. The content format follows Table A2.
>
> |  Method  |    UA     |    TA     |
> |:--------:|:---------:|:---------:|
> |    GA    |   81.53   |   21.33   |
> | GA+$m_s$ | **85.18** | **46.72** |
>
> > [R4] Thudi, Anvith, et al. "Unrolling sgd: Understanding factors influencing machine unlearning." 2022 IEEE 7th European Symposium on Security and Privacy (EuroS&P). IEEE, 2022
>
> **Response to W3 regarding other weight saliency methods in continual learning literature:**
>
> We appreciate the suggestion to explore the continual learning corpus. However, it seems that in this context, the notion of saliency aligns more with the **data** level rather than the **weight** level. We are enthusiastic about broadening our comprehension, particularly regarding any existing literature on weight saliency within continual learning. We would greatly appreciate additional guidance or insights from Reviewer TAAE in this regard.
>
> **Response to W4 regarding limitations of the proposed method:**
>
> Thanks for raising this question. We will further strengthen the Limitation section (Appendix C) in the supplement.  Specifically, we will elaborate on the potential limitations below
>
> 1. The characteristics of language tasks, as opposed to vision tasks, suggest that saliency units in LMs may encompass more complex structures such as attention heads or feedforward network weights/columns. In addition, the saliency evaluation loss should also be carefully designed for language tasks. Thus, the efficacy of our proposed methods on LLMs requires additional exploration.
> 2. The heuristic-based approach for saliency threshold selection has demonstrated empirical success; nevertheless, an automated mechanism for optimal threshold determination remains a goal for future research, particularly for application to novel model architectures.
>
> ---
>
> We hope our responses sufficiently address the concerns raised and contribute to the constructive dialogue around our submission. We are looking forward to hearing from Reviewer TAAE for further discussion.

---

> ### Author Response · Authors · 2023-11-22
> **Reminder on follow-up discussion (one day left before rebuttal ends)**
>
> Dear Reviewer TAAE,
>
> We sincerely appreciate the time and effort you've invested in reviewing our manuscript. Your expertise and dedication to the review process are truly invaluable to us.
>
> As the discussion phase draws to a close in one day, we kindly request you to provide your feedback on our rebuttal. Your insights are of immense importance to us, and we look forward to any additional comments you may have. If our replies have addressed your concerns, we would be thankful for your recognition of this. Should there be any points requiring further discussion or explanation, please feel free to contact us. Furthermore, we are fully prepared to engage more to enhance our work during this pivotal stage.
>
> Best regards,
>
> Authors

---

### Official Review · Reviewer_XDsn · 2023-11-07

**Soundness:** 3 good
**Presentation:** 3 good
**Contribution:** 3 good
**Rating:** 8
**Confidence:** 2

**Summary:**

This paper introduces a novel concept of 'saliency unlearning' (SalUn) to address the challenges of machine unlearning (MU) in the context of data protection and trustworthy machine learning. The paper critiques the instability and adaptability issues of current MU methods and proposes a weight saliency-based approach to enhance MU performance in both image classification and generation tasks. The authors provide experimental comparisons with existing MU methods, demonstrating SalUn's effectiveness, especially in preventing harmful content generation in diffusion models. The paper is well-structured, with comprehensive experiments validating the proposed method.

**Strengths:**

+ SalUn introduces a novel gradient-based weight saliency approach, a significant departure from existing MU methods.
+ Demonstrates SalUn's practical utility in preventing the generation of harmful content, an important aspect for the deployment of generative models.

**Weaknesses:**

- The authors could enhance the paper by providing a more thorough analysis of the potential limitations or scenarios where the SalUn method may not perform optimally.

**Questions:**

1. Can the authors elaborate on how SalUn would handle incremental unlearning scenarios where data points are continuously added and removed?
2. What are the computational costs associated with SalUn compared to traditional retraining methods, especially for very large datasets?
3. Include a discussion on the potential limitations of SalUn, such as scenarios where it may fail or be less effective, to provide a balanced view.

---

> ### Author Response · Authors · 2023-11-19
> **Response to Reviewer XDsn**
>
> We extend our sincerest thanks to Reviewer XDsn for the encouraging comments on the contributions and presentation quality of our paper. We are equally appreciative of the constructive questions posed. Below, we provide our responses to the comments, denoted by **[W]** for weaknesses and **[Q]** for questions.
>
> **Response to W1 and Q3 on limitation discussion:**
>
> Thank you for bringing up this question. We will add more discussions on the potential limitations of our technique in the main texts and enhance the Limitation section (Appendix C) in the supplement to provide a more comprehensive understanding of potential limitations. In particular, we will elaborate on the following points:
> 1. Language tasks exhibit different characteristics compared to vision tasks, which suggests that saliency units in Language Models (LMs) may involve more complex structures, such as attention heads or feedforward network weights/columns. In addition, the design of saliency evaluation loss should also be approached with careful consideration for language tasks. Therefore, the effectiveness of our proposed methods on LMs warrants further investigation.
> 2. While our heuristics-based approach for saliency threshold selection has shown empirical success, the pursuit of an automated mechanism to determine optimal thresholds remains a goal for future research. This is especially important when considering the application of our methods to novel model architectures.
>
>
>
> **Response to Q1 on incremental unlearning:**
>
> Thanks for raising this question. Based on your suggestion, we conducted iterative unlearning experiments (see **Table R3** below) by incrementally forgetting 10% of the data over five iterations (50% of data in total), *i.e.*, for each iteration the forgetting set $(D_f)$ is 10% of the whole dataset, given ResNet-18 on the CIFAR-10 dataset. We evaluate the unlearning performance of our method by comparing it with the gold standard, retraining. Additionally, we assess its performance in comparison to an unlearning baseline, FT (fine-tuning). The results are presented in the following table. Notably, even as data points are progressively forgotten, our method demonstrates a consistently minimal performance gap with Retrain, as evidenced by the smallest value in the "Avg. Gap" column.
>
> **Table R3**: Performance of incremental unlearning on ResNet-18, pre-trained on CIFAR-10, for random data forgetting. The content format follows Table 1.
> |  Iteration   | Method  |  UA   |   RA   |  TA   |  MIA  | Avg. Gap |
> |:------------:|:------- |:-----:|:------:|:-----:|:-----:|:--------:|
> |      1       | Retrain | 5.24  | 100.00 | 94.26 | 12.88 |   0.00   |
> | (Forget 10%) | Ours    | 4.69  | 99.52  | 93.35 | 14.40 | **0.87** |
> |              | FT      | 11.38 | 91.46  | 86.97 | 17.69 |   6.70   |
> |      2       | Retrain | 5.31  | 100.00 | 94.10 | 13.30 |   0.00   |
> | (Forget 20%) | Ours    | 6.60  | 98.52  | 91.72 | 14.38 | **1.56** |
> |              | FT      | 10.60 | 97.27  | 87.81 | 19.42 |   5.11   |
> |      3       | Retrain | 6.64  | 100.00 | 92.78 | 14.60 |   0.00   |
> | (Forget 30%) | Ours    | 7.47  | 97.75  | 91.03 | 15.89 | **1.53** |
> |              | FT      | 10.56 | 96.89  | 85.66 | 20.38 |   4.98   |
> |      4       | Retrain | 7.01  | 100.00 | 92.52 | 18.37 |   0.00   |
> | (Forget 40%) | Ours    | 8.49  | 99.21  | 91.73 | 20.64 | **1.33** |
> |              | FT      | 8.82  | 97.64  | 85.42 | 18.62 |   2.88   |
> |      5       | Retrain | 7.91  | 100.00 | 91.72 | 19.29 |   0.00   |
> | (Forget 50%) | Ours    | 8.76  | 98.91  | 90.28 | 22.58 | **1.69** |
> |              | FT      | 9.00  | 96.87  | 84.29 | 18.78 |   3.02   |
>
> **Response to Q2 on computation costs:**
>
> Thank you for your question. In comparison to Retrain, SalUn exhibits significantly improved computational efficiency. This efficiency stems from the fact that, unlike Retrain, which necessitates retraining the model from scratch after removing forgotten data points, SalUn's computational process primarily involves two key components: The gradient-based weight saliency map, and a fast approximate unlearning procedure. The saliency map, as outlined in Eq. 4, provides a plug-in guidance on which variables should be updated during the unlearning process. As a result, SalUn's efficiency aligns with that of “fast” approximate unlearning methods. We have quantified this efficiency in terms of runtime efficiency (RTE), as presented in relevant tables, such as Table 1 and A1. It is worth noting that SalUn's efficiency remains applicable even when dealing with a larger dataset. For instance, in scenarios involving Stable Diffusion, retraining is often not a feasible solution for removing the influence of NSFW concepts. This makes SalUn a practical and efficient choice.

---

> ### Author Response · Authors · 2023-11-22
> **Reminder on follow-up discussion (one day left before rebuttal ends)**
>
> Dear Reviewer XDsn,
>
> We sincerely appreciate the time and effort you've invested in reviewing our manuscript. Your expertise and dedication to the review process are truly invaluable to us.
>
> As the discussion phase draws to a close in one day, we kindly request you to provide your feedback on our rebuttal. Your insights are of immense importance to us, and we look forward to any additional comments you may have. If our replies have addressed your concerns, we would be thankful for your recognition of this. Should there be any points requiring further discussion or explanation, please feel free to contact us. Furthermore, we are fully prepared to engage more to enhance our work during this pivotal stage.
>
> Best regards,
>
> Authors

---

### Official Review · Reviewer_miKP · 2023-11-08

**Soundness:** 3 good
**Presentation:** 4 excellent
**Contribution:** 4 excellent
**Rating:** 8
**Confidence:** 3

**Summary:**

Inspired by modular ML approaches such as weight sparsity, this work introduces a Machine Unlearning (MU) approach for image classification and generation that creates weight saliency maps to identify weights that need to be unlearned and weights to keep unchanged. The map coefficients are set to 1 when the magnitude of gradient of the forgetting loss is higher than a threshold (set heuristically to the median of the gradient of the forgetting loss). A first assessment of classical MU approaches on CIFAR10 motivates the need for a better approach, then the experiments demonstrate the superiority of the proposed approach SAGUN on CIFAR10, CIFAR100, SVHN, ImageNette using 4 relevant metrics for machine unlearning. Authors also provide quantitative and qualitative evaluation of their approach applied to reduce the number of nudity image generation, showing large improvements over the previous works on generative machine unlearning ESD and FMN.

**Strengths:**

* Clearly written and motivated work.
* Significance: the task is very relevant: machine unlearning for generative models, a very important topic to prevent harmful content generation, was lacking an effective solution.
* The approach is simple, effective, modular, and original to my knowledge.
* The experimental section presents extensive results, qualitative and quantitative.

**Weaknesses:**

* The main table of numerical results is on CIFAR10, with close to 100% RA accuracy, so makes it difficult to compare approaches results. The gap with concurrent approaches reduces a bit on CIFAR100. It would be interesting to see a comparison on a dataset with larger images to make sure the approach scales effectively.
* Some choices lack justifications, for instance, in the motivation figure, IU is picked as an example, and in the approach section, the forgetting loss is using the Gradient Ascent (GA) one. We see later that Table A1 presents results with the combination of different approaches.
* There are not many parameters to tune, but I did not see an ablation on the choice of alpha in (7). What are the results with alpha =0? How was the number of steps chosen in the different cases?
* No code available.

**Questions:**

For my main concerns, see the Weakness section.

Specific asks for Fig 6:
a) given the small size of the figure, one of the image is looking like a nudity example, it is only with a high zoom that one can see the person is not actually naked. I would suggest to also mask this image (P6, ESD).
b) I would suggest blurring the faces of identifiable persons here as it does not bring anything more to the scientific content of the paper.
Similarly, in the text, I don't think giving an example of a nudity prompt is necessary.

Minor comments:

* The approaches behind the two generative unlearning previous works could be discussed a bit more in the related work section.
* Maybe consider citing the Gradient surgery paper https://arxiv.org/pdf/2307.04550.pdf in the related work section.

* Remove the "as we can see" (multiple occurrences) to save space
* Maybe revise sentence "existing MU methods tend to either over-forget, resulting in poor generation quality ... (e.g. GA, RL) or under-forget: I don't see the causality link here, with very similar results of the GA and RL for both sets of classes.
* Page 8 : "which contradicts the results obtained by retrain.." -> I did not understand why it did contradict the Retrain results, maybe add a little detail here.

---

> ### Author Response · Authors · 2023-11-20
> **Response to Reviewer miKP (1/2)**
>
> We extend our sincerest thanks to Reviewer miKP for the meticulous examination of our work, including the supplement materials, and for the encouraging comments on the contributions and presentation quality of our paper. We also appreciate the posted constructive questions. Below, we provide our responses to the comments, denoted by **[W]** for weaknesses and **[Q]** for questions.
>
> **Response to W1 regarding nearly 100% RA accuracy:**
>
> Thank you for your question. To provide a more comprehensive answer, let us revisit the evaluation metrics in **MU for image classification** of Sec. 3. When assessing MU, it is essential to consider three key aspects:
>
> 1. **Unlearning fidelity** (UA, MIA in classification, UA in generation): This aspect assesses whether the model can successfully unlearn previously learned information.
> 2. **Preserved generalization or generation ability** (RA, TA in classification, FID in generation): This measures the model's ability to maintain comparable "accuracy" or performance on tasks unrelated to the unlearning process.
> 3. **Run-time efficiency** (RTE).
>
> Given the 2nd criterion, **RA** and **TA** should be considered together because RA evaluates performance on the remaining training dataset, while TA evaluates performance on the testing dataset. Since the remaining dataset has been previously seen by the pre-trained model, it is not surprising that RA could maintain at a relatively high level, especially in the scenario of random data forgetting. Thus, RA can be regarded as a sanity checker of a “good” unlearning method without losing accuracy in the remaining training set. However, it is important to note that despite most of RA being close to 100% in Table 1, this does not imply that different unlearning methods perform equally well in the 2nd criterion mentioned above (preserved generalization/generation ability). It is evident from Table 1 that TA exhibits a higher rate of change when compared to RA across various unlearning methods.
>
> Based on the reviewer’s suggestion, we have conducted an **additional experiment** on the TinyImageNet dataset (see **Table R1**) with a larger image size of 64 * 64 than CIFAR-10/100. Due to time constraints, we focused on evaluating our approach against the fine-tuning (FT) and gradient ascent (GA) unlearning baselines. We also included the performance of the Retrain method for the sake of comparison. As evident from the results, both FT and GA fall short of matching the performance of our method, SalUn. This underperformance is notably reflected in the larger performance gap (Ave. Gap) when compared to the performance of Retrain. To delve deeper into the specifics, FT appears to excel in preserving generalization ability (criterion 2), yet it struggles when it comes to the unlearning aspect (criterion 1). Conversely, GA exhibits effectiveness in the unlearning process, but at the cost of significantly diminishing both RA and TA.
>
> **Table R1**: MU Performance on ResNet-18, pre-trained on Tiny ImageNet, for forgetting within a single class. The table format follows Table A2.
> |         |  UA  |  RA   |  TA   | MIA  | Avg. Gap |
> | :-----: | :--: | :---: | :---: | :--: | :-------: |
> | Retrain | 100  | 98.66 | 65.07 | 100  |     0     |
> |   FT    | 24.6 | 99.78 | 64.41 | 49.6 |   31.89   |
> |   GA    | 86.8 | 90.35 | 58.35 | 92.2 |   9.01    |
> |  Ours   | 87.6 | 99.42 | 65.35 | 100  |   3.36    |
>
>
>
>
> **Response to W2 on the lack of justifications for loss choices:**
>
> Thank you for raising this question. We believe there might exist a misunderstanding due to our unclear presentation in certain parts. We apologize for the confusion we have caused. Here we further clarify our loss choices.
>
> 1. The forgetting loss in Eq. 3 is proposed for the computation of the saliency map, and it is given by the training loss over the forgetting dataset. Gradient ascent (GA) is the method name rather than the loss name. The rationale behind using Eq. 3 is inspired by the GA method as it favors unlearning fidelity performance [Sec. 5, R1]. We then use the forgetting loss of GA to identify the subset of weights salient to unlearning.
>
> 2. After the determination of the saliency map, we employ a more advanced unlearning loss, tailored for not only unlearning fidelity but also preserved generalization/generation ability. A saliency map is a plug-in prior to which variables should be updated during unlearning. In other words, we used our proposed unlearning loss + saliency map (calculated by step 1 above). Table A1 demonstrates the impact of integrating the discovered saliency map into various unlearning methods.
>
> > [R1] Thudi, Anvith, et al. "Unrolling sgd: Understanding factors influencing machine unlearning." 2022 IEEE 7th European Symposium on Security and Privacy (EuroS&P). IEEE, 2022.

---

> ### Author Response · Authors · 2023-11-20
> **Response to Reviewer miKP (2/2)**
>
> **Response to W3 regarding the choice of hyperparameters in Eq. 7:**
>
> We appreciate your suggestion. We provide an additional ablation study (see **Table R2** below) to showcase the influence of $\alpha$ on the unlearning performance in image generation using Stable Diffusion (SD) on ImageNette. The experimental setup aligns with Table 2. To analyze the impact of the variable $\alpha$ on both unlearning fidelity and preserved generation ability, we calculate FID scores for both the forgetting class condition $(C_f)$ and the remaining class condition $(C_r)$.
>
> We aim for the generated images corresponding to the forgetting class not to belong anymore, meaning a higher UA closer to 100% is desirable. Simultaneously, we seek higher quality in the generated images, indicating a preference for a lower FID score. Further, we aim to minimize the discrepancy in the generated image quality between the forgetting and the remaining classes, signifying that a closer alignment between the FID scores of generated images is conditioned on the forgetting class and the remaining class. This ensures the proximity to the retrained model, as shown in Fig. 3. As illustrated in Table R2 when the parameter $\alpha$ is varied within the range ${0.05, 0.5, 5}$, the unlearning task consistently maintains successful, with 100% UA (unlearning accuracy) for the generated images under the forgetting class condition $(C_f)$. However, it is worth noting that the choice of $\alpha$ has a discernible impact on the quality of the generated images. This quality is quantified through the measurement of FID scores under both $C_f$ and the remaining class condition $(C_r)$, denoted as FID $(C_f)$ and FID $(C_r)$, respectively. When $\alpha$ increases,  FID $(C_r)$ decreases. This is not surprising since there will be more penalization on the MSE loss under the remaining dataset in Eq. (7). In turn, this will hamper the forgetting term in Eq. (7), causing a higher FID $(C_f)$. Through the lens of aligned image generation quality regardless of $C_r$ or $C_f$, we then choose $\alpha = 0.5$  in our experiments.
>
> In an extreme case of $\alpha=0$, we will face the issue of over-unlearning. This significantly compromises image generation quality. The FID scores under both the forgetting and remaining conditions are extremely high (~300), indicating the malfunctioning of the diffusion model in image generation.
>
> Further, we evaluate the UA for diffusion models every epoch by generating a few samples under the forgetting condition, and stopping the unlearning process once the condition is successfully unlearned, *i.e.*, whether the UA approaching 100%. As Appendix B.1 mentioned, the empirical value for SD lands on 5 epochs.
>
> **Table R2:** Performance of class-wise forgetting on Imagenette using SD with different $\alpha$. The setting follows Table 2.
> | $\alpha$ | FID $(C_f)$ | FID $(C_r)$ | UA |
> |:--------:|:------:|:----------:|:----------:|
> |   0.05    |    2.35    |    7.72    | 100 |
> |   0.50   |    3.03    |    2.54    | 100 |
> |   5.00   |    5.07    |    0.84    | 100 |
>
> **Response to W4 regarding code availability:**
>
> We acknowledge your suggestion and commit to making our code publicly available upon revision to facilitate transparency and reproducibility.
>
> **Response to Q regarding minor comments on figure presentation, reference, and wording:**
>
> Thank you very much for the very careful reading. We are committed to enhancing the clarity of our presentations, both in our written content and visual presentations, in line with your suggestions. Additionally, we will ensure that we appropriately cite the suggested reference and provide more in-depth discussions regarding the two generative unlearning methods mentioned in the related work. Thank you for your constructive input, and we look forward to further improving the quality of our work.

---

> ### Author Response · Authors · 2023-11-22
> **Reminder on follow-up discussion (one day left before rebuttal ends)**
>
> Dear Reviewer miKP,
>
> We sincerely appreciate the time and effort you've invested in reviewing our manuscript. Your expertise and dedication to the review process are truly invaluable to us.
>
> As the discussion phase draws to a close in one day, we kindly request you to provide your feedback on our rebuttal. Your insights are of immense importance to us, and we look forward to any additional comments you may have. If our replies have addressed your concerns, we would be thankful for your recognition of this. Should there be any points requiring further discussion or explanation, please feel free to contact us. Furthermore, we are fully prepared to engage more to enhance our work during this pivotal stage.
>
> Best regards,
>
> Authors

---

> > ### Comment · Reviewer_miKP · 2023-11-22
> > **Satisfied with Rebuttal**
> >
> > Hello, I read the other reviews and authors responses. All reviewers rated the paper as above the acceptance threshold. I am satisfied in the answers overall and in particular to my concerns and appreciate the extra experiment on larger images. Also mitigations to Fig 6, and removal of problematic prompts are appreciated. My rating does not change, I recommend acceptance.

---

> > > ### Author Response · Authors · 2023-11-22
> > > **Thank you!**
> > >
> > > Dear Reviewer miKP,
> > >
> > > We sincerely appreciate your prompt response and are pleased that you found our additional comments beneficial. We're thrilled that your score will be maintained.
> > >
> > > Thank you again for your valuable input in enhancing our submission.
> > >
> > > Best regards,
> > >
> > > Authors

---

### Meta-Review · Area_Chair_Jp39 · 2023-12-08

**Metareview:**

This paper proposes a novel machine unlearning approach called "Saliency Unlearning" (SalUn). The goal is to improve the accuracy, stability, and cross-domain applicability of unlearning data from AI models. SalUn creates weight saliency maps to identify weights that need to be unlearned and weights to keep unchanged, achieving high unlearning accuracy even in challenging scenarios like random data forgetting and harmful image prevention. The reviewers have praise the clarity of the presentation and convincing experimental results. The tackled problem is very timely and goes beyond the traditional safeguards that are put in place : prompt engineering or prompt filtering. Because of the above, I recommend accepting this paper as a spotlight.

**Justification For Why Not Higher Score:**

The current presentation of the work grants a spotlight presentation. In order to go above, I think the current approach should show a very general applicability. For instance it should be generalized to other generative models, and in particular LLMs.

**Justification For Why Not Lower Score:**

This work is well presented and the proposed methods works well. The tackled problem is important and the proposed solution is technically sound and novel.

---

### Decision · Program_Chairs · 2024-01-16

Accept (spotlight)